# Deep Reinforcement Learning for Industrial Insertion Tasks with Visual Inputs and Natural Reward Signals

**Gerrit Schoettler** [* 1]   **Ashvin Nair** [* 2]   **Jianlan Luo** [2]   **Shikhar Bahl** [2]
**Juan Aparicio Ojea** [1]   **Eugen Solowjow** [1]   **Sergey Levine** [2]

## Abstract

Connector insertion and many other tasks commonly found in modern manufacturing settings involve complex contact dynamics and friction. Since it is difficult to capture related physical effects with first-order modeling, traditional control methodologies often result in brittle and inaccurate controllers, which have to be manually tuned. Reinforcement learning (RL) methods have been demonstrated to be capable of learning controllers in such environments from autonomous interaction with the environment, but running RL algorithms in the real world poses sample efficiency and safety challenges. Moreover, in practical real-world settings we cannot assume access to perfect state information or dense reward signals. In this paper we consider a variety of difficult industrial insertion tasks with visual inputs and different natural reward specifications, namely sparse rewards and goal images. We show that methods that combine RL with prior information, such as classical controllers or demonstrations, can solve these tasks directly by real-world interaction.

## 1. Introduction

There is a gap between today's robotic automation capability and modern industrial tasks driven by the required adaptability to increasing production flexibility. While traditional control methods, such as PID regulators, are heavily employed to automate many tasks in the context of positioning, tasks that require significant adaptability or tight visual perception-control loops are often beyond the capabilities of traditional control methods and hence left to humans. This is because traditional control methods often struggle to accurately model realistic elements of real-world dynamics, such as friction and contacts. Alternatively, reinforcement

learning (RL) offers a different solution: instead of attempting to create perfect models, learn a policy that attempts to solve the task even in the presence of poor models.

While promising, deep RL has thus far not seen wide adoption in the automation community due to several practical obstacles. Sample efficiency is one obstacle: tasks must be completed without excessive interaction time or wear and tear on the robot. Progress in recent years on developing better RL algorithms have led to significantly better sample efficiency, even in dynamically complicated tasks. Another major, often underappreciated, obstacle is goal specification: while prior work in RL assumes a reward signal to optimize, it is often carefully shaped to allow the system to learn. Obtaining such dense reward signals can be a significant challenge, as one must additionally build a perception system that allows computing dense rewards on state representations. Also, shaping a reward signal so that an agent can learn from it is a manual process that requires a human in the loop. Instead, we would ideally be able to learn in the presence of rewards that are more natural to specify. How can we allow robots to autonomously perform complex tasks without significant engineering effort to design perception and reward systems? The first method we consider is using end-to-end learning from vision, using images for state information and goal images for reward specification. Using goal images is not fully general, but can successfully represent tasks when the task is to reach a final desired state. Allowing goal specification through goal images is convenient and eases the requirement of a human in the training loop specifying dense rewards. Furthermore, using images

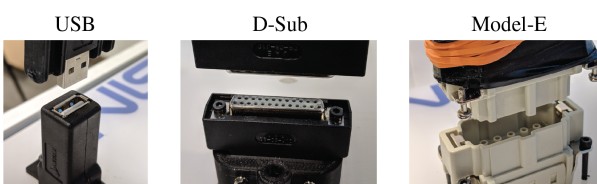

USB · D-Sub · Model-E

*Figure 1.* We train an agent directly in the real world to solve connector insertion tasks that involve contacts and tight tolerances from convenient reward signals such as pixel distance to a goal image or a sparse electrical signal.

---

[*]Equal contribution. [1]Siemens Corporation, Berkeley, USA [2]Berkeley AI Research, University of California, Berkeley, Computer Science. Correspondence to: Ashvin Nair <anair17@berkeley.edu>.

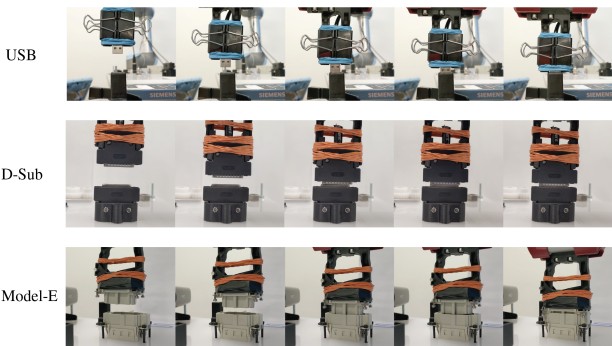

*Figure 2.* The three image sequences show rollouts from learned policies that successfully complete the insertion tasks. For each task, a separate policy is trained.

for state information can allow the agent to learn control policies that are robust to sensor and actuator noise, as the agent can infer how to complete the task visually.

Secondly, we consider learning from sparse rewards. Sparse rewards can often be obtained conveniently, for instance by a human labeller or by natural mechanical or electrical systems. In many electronic assembly tasks, which we consider here, we can directly detect whether the electronics are functional and use that signal as a reward. Learning from sparse rewards poses a challenge, as exploration with sparse reward signals is difficult, but by using sufficient prior information about the task, one can overcome this challenge. Specifically, to handle this challenge we use residual RL (Johannink et al., 2019; Silver et al., 2018) which learns a parametric policy on top of a fixed, hand-specified policy in order to provide safer and more sample efficient exploration.

In this work, we show that we can successfully complete real-world tight tolerance assembly tasks, such as inserting USB connectors and other connectors into an electrical board, using deep RL with reward signals that are convenient to specify. In particular, we show that we can learn from only a sparse reward of electrically detecting whether the USB adapter is plugged in. Also, we demonstrate that our approach can learn the same insertions only from goal images. These reward signals require no extra engineering and are natural to specify for many tasks. Beyond showing the feasibility of RL to solve these tasks, we comprehensively evaluate algorithms across three tasks and evaluate their robustness to noise.

## 2. Related Work

Learning has been applied previously in a variety of robotics contexts. Different forms of learning have enabled autonomous driving (Pomerleau, 1989), biped locomotion (Nakanishi et al., 2004), block stacking (Deisenroth et al., 2011), grasping (Pinto & Gupta, 2016), and navigation

(Giusti et al., 2015; Pathak et al., 2018). Among these methods, many involve reinforcement learning, where an agent learns to perform a task by maximizing a reward signal. Reinforcement learning algorithms have been developed and applied to teach robots to perform tasks such as balancing a robot (Deisenroth & Rasmussen, 2011), playing ping-pong (Peters et al., 2010) and baseball (Peters & Schaal, 2008). The use of large function approximators, such as neural networks, in RL has further broadened the generality of RL (Mnih et al., 2013). Such techniques, called "deep" RL, have further allowed robots to perform fine-grained manipulation tasks from vision (Levine et al., 2016), open doors (Gu et al., 2016), score a hockey puck (Chebotar et al., 2017), and grasp objects (Kalashnikov et al., 2018). In this work we further explore solving real-world robotics tasks using RL.

Many RL algorithms introduce prior information about the specific task to be solved through various means such as reward shaping (Ng et al., 1999), incorporating a trajectory planner (Thomas et al., 2018; Eruhimov & Meeussen, 2011; Mayton et al., 2010), learning classifiers between goals and non-goals (Ho & Ermon, 2016; Pinto & Gupta, 2016; Levine et al., 2017). These methods require access to various goal states to build a robust classifier, which might be difficult to collect in assembly as there is often only one goal image possible. Reward shaping can become arbitrarily difficult as the complexity of the task increases. For complex assembly tasks, trajectory planners require a host of information about objects and geometries which can be difficult to provide.

Mainly, previous work on incorporating prior information has focused on using demonstrations either to initialize a policy (Peters & Schaal, 2008; Kober & Peter, 2008), infer reward functions using inverse reinforcement learning (Finn et al., 2016; Abbeel & Ng, 2004; Ziebart et al., 2008; Rhinehart & Kitani, 2017; Fu et al., 2018) or to improve the policy throughout the learning procedure (Hester et al., 2018; Nair et al., 2018; Rajeswaran et al., 2018; Večerík et al., 2017). These methods require multiple demonstrations, which can be difficult to collect, especially for assembly tasks. More recently, manually specifying a policy and learning the residual task has been proposed (Johannink et al., 2019; Silver et al., 2018). In this work we evaluate both residual RL and combining RL with learning from demonstrations (LfD).

Previous work has also tackled high precision assembly tasks, especially insertion-type tasks. One line of work focuses on obtaining high dimensional observations, including geometry, forces, joint positions and velocities (Li et al., 2014; Tamar et al., 2017; Inoue et al., 2017; Luo et al., 2019), but this information is not easily procured, increasing complexity of the experiments and the supervision required to collect the data. Other work relies on external trajectory planning or very high precision control (Inoue et al., 2017;

Tamar et al., 2017), but this can be brittle to error in other components of the system, such as perception. We show how our method not only solves insertion tasks with much less information about the environment, it also does so under noisy conditions.

## 3. Electric Connector Plug Insertion Tasks

In this work we empirically evaluate learning methods on a set of electric connector assembly tasks, pictured in Fig. 1. Connector plug insertions are difficult for two reasons. First, the robot must be very precise in lining up the plug with its socket. As we show in our experiments, errors as small as $\pm 1$ mm can lead to consistent failure. Second, there is significant friction when the connector plug touches the socket, and the robot must learn to apply sufficient force in order to successfully insert the plug. In our experiments, we use a 7 degrees of freedom Sawyer robot with end-effector control, meaning that the action signal $u_t$ can be interpreted as the relative end-effector movement in Cartesian coordinates. The robot's internal controller is illustrated in Fig. 3.

To comprehensively evaluate connector assembly tasks, we repeat our experiments on a variety of connectors. Each connector offers a different challenge in terms of required precision and force to overcome friction. We chose to benchmark the controllers performance on the insertion of a USB connector, a U-Sub connector, and a waterproof Model-E connector manufactured by MISUMI. All the explored use cases were part of the IROS 2017 Robotic Grasping and Manipulation Competition (Falco et al., 2018), included as part of a task board developed by NIST to benchmark the performance of assembly robots.

### 3.1. Adapters

In the following we describe the used adapters, USB, D-Sub and Model-E, in order by the observed difficulty level of the insertion task.

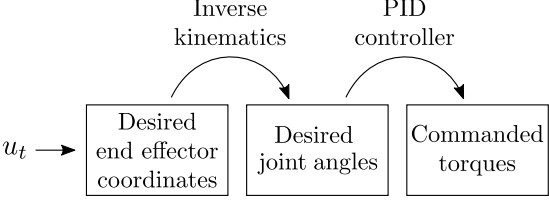

*Figure 3.* Illustration of the robot's underlying cascaded control scheme. The actions $u_t$ are computed at a frequency of up to 10 Hz, desired joint angles are obtained by solving an inverse kinematics problem, and a joint-space impedance controller with anti-windup PID control commands actuator torques at 1000 Hz.

### 3.1.1. USB

The USB connector is a ubiquitous, widely-used connector and offers a challenging insertion task. Because the adapter becomes smoother and therefore easier to insert over time due to wear and tear, we periodically replace the adapter. Of the three tested adapters, the USB adapter is the easiest.

### 3.1.2. D-SUB

Inserting this adapter requires aligning several pins correctly, and is therefore more sensitive than inserting the USB adapter. It also requires more downward force due to a tighter fit.

### 3.1.3. MODEL-E

This adapter is the most difficult of the three tested connectors as it contains several edges and grooves to align and requires significant downward force to successfully insert the part.

### 3.2. Experimental Settings

We consider three settings in our experiments in order to evaluate how plausible it is to solve these tasks with more convenient state representations and reward functions and to evaluate the performance different algorithms as the setting is modified.

### 3.2.1. VISUAL

In this experiment, we evaluate whether the RL algorithms can learn to perform the connector assembly tasks from vision without having access to state information. The state provided to the learned policy is a $32 \times 32$ image converted to grayscale. For goal specification, we use a goal image, avoiding the need for state information to compute rewards. The reward is the pixelwise L1 distance to the given goal image. Being able to learn from such a setup is compelling as it does not require any extra state estimation and many tasks can be specified easily by a goal image.

### 3.2.2. ELECTRICAL (SPARSE)

In this experiment, the reward is obtained by directly measuring whether the connection is alive and transmitting:

$$r = \begin{cases} 1, & \text{if insertion signal detected} \\ 0, & \text{else.} \end{cases} \tag{1}$$

This is the exact true reward for the task of connecting a cable, and can be naturally obtained in many manufacturing systems. As state, the robot is given the Cartesian coordinates of the end-effector $x_t$ and the vertical force $f_z$ that is acting on the end-effector. As we could only automatically detect the USB connection thus far, we only include the

USB adapter for the sparse experiments.

### 3.2.3. DENSE

In this experiment, the robot receives a manually shaped reward based on distance to the target. We use the reward function

$$r_t = -\alpha \cdot \|x_t - x^*\|_1 - \frac{\beta}{(\|x_t - x^*\|_2 + \varepsilon)} - \varphi \cdot f_z, \quad (2)$$

where $0 < \varepsilon \ll 1$. The hyperparameters are set to $\alpha = 100$, $\beta = 0.002$, and $\varphi = 0.1$. When an insertion is indicated through a distance measurement, the sign of the force term flips, so that $\varphi = -0.1$. This rewards the agent for pressing down after an insertion and showed to improve the learning process.

## 4. Methods

To solve the connector insertion tasks, we consider and evaluate a variety of reinforcement learning algorithms.

### 4.1. Preliminaries

In a Markov decision process (MDP), an agent at every time step is at state $s_t \in \mathcal{S}$, takes actions $u_t \in \mathcal{U}$, receives a reward $r_t \in \mathbb{R}$, and the state evolves according to environment transition dynamics $p(s_{t+1}|s_t, u_t)$. The goal of reinforcement learning is to choose actions $u_t \sim \pi(u_t|s_t)$ to maximize the expected returns $\mathbb{E}[\sum_{t=0}^{H} \gamma^t r_t]$ where $H$ is the horizon and $\gamma$ is a discount factor. The policy $\pi(u_t|s_t)$ is often chosen to be an expressive parametric function approximator, such as a neural network, as we use in this work.

### 4.2. Efficient Off-Policy Reinforcement Learning

One class of RL methods additionally estimates the expected discounted return after taking action $u$ from state $s$, the Q-value $Q(s, u)$. Q-values can be recursively defined with the Bellman equation:

$$Q(s_t, u_t) = \mathbb{E}_{s_{t+1}}[r_t + \gamma \max_{u_{t+1}} Q(s_{t+1}, u_{t+1})] \quad (3)$$

and learned from off-policy transitions $(s_t, u_t, r_t, s_{t+1})$. Because we are interested in sample-efficient real-world learning, we use such RL algorithms that can take advantage of off-policy data.

For control with continuous actions, computing the required maximum in the Bellman equation is difficult. Continuous control algorithms such as deep deterministic policy gradients (DDPG) (Lillicrap et al., 2016) additionally learn a policy $\pi_\theta(u_t|s_t)$ to approximately choose the maximizing action. In this paper we specifically consider two related reinforcement learning algorithms that lend themselves well

to real-world learning as they are sample efficient, stable, and require little hyperparameter tuning.

### 4.2.1. TWIN DELAYED DEEP DETERMINISTIC POLICY GRADIENTS (TD3)

Like DDPG, TD3 optimizes a deterministic policy (Fujimoto et al., 2018) but uses two Q-function approximators to reduce value overestimation (Van Hasselt et al., 2016) and delayed policy updates to stabilize training.

### 4.2.2. SOFT ACTOR CRITIC (SAC)

SAC is an off-policy value-based reinforcement learning method based on the maximum entropy reinforcement learning framework with a stochastic policy (Haarnoja et al., 2018).

We used the implementation of these RL algorithms publicly available at `rlkit` (Pong et al., 2018).

### 4.3. Residual Reinforcement Learning

Instead of randomly exploring from scratch, we can inject prior information into an RL algorithm in order to speed up the training process, as well as minimize unsafe exploration behavior. In residual RL, actions $u$ are chosen by additively combining a fixed policy $\pi_H(s)$ with a parametric policy $\pi_\theta(u_t|s_t)$:

$$u = \pi_H(s) + \pi_\theta(s). \quad (4)$$

The parameters $\theta$ can be learned using any RL algorithm. In this work, we evaluate both SAC and TD3, explained in the previous section. The residual RL implementation that we use in our experiments is summarized in Algorithm 1.

---

**Algorithm 1** Residual reinforcement learning

**Require:** policy $\pi_\theta$, hand-engineered controller $\pi_H$.
1: **for** $n = 0, ..., N - 1$ episodes **do**
2:     Sample initial state $s_0 \sim E$.
3:     **for** $t = 0, ..., H - 1$ steps **do**
4:         Get policy action $u_t \sim \pi_\theta(u_t|s_t)$.
5:         Get action to execute $u'_t = u_t + \pi_H(s_t)$.
6:         Get next state $s_{t+1} \sim p(\cdot \mid s_t, u'_t)$.
7:         Store $(s_t, u_t, s_{t+1})$ into replay buffer $\mathcal{R}$.
8:         Sample set of transitions $(s, u, s') \sim \mathcal{R}$.
9:         Optimize $\theta$ using RL with sampled transitions.
10:     **end for**
11: **end for**

---

A simple P-controller serves as the hand-designed controller $\pi_H$ of our experiments. The P-controller operates in Cartesian space and calculates the current control action by

$$u_t = -k_p \cdot (x_t - x^*), \quad (5)$$

where $x^*$ denotes the commanded goal location. As control

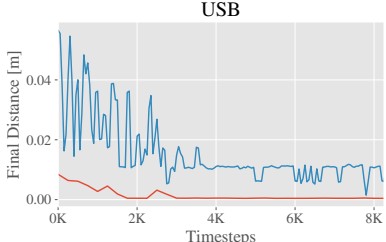 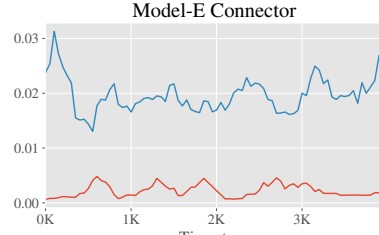 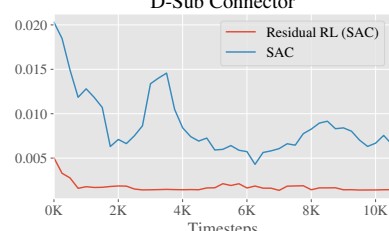

*Figure 4.* Plots of the final mean distance to the goal during the state-based training. Final distances greater than $0.01\,\mathrm{m}$ indicate unsuccessful insertions. Here, the residual RL approach performs noticeably better than pure RL and is often able to solve the task during the exploration in the early stages of the training.

gains we use $k_p = [\,1,\,1,\,0.3\,]$. This P-controller quickly centers the end effector above the goal position and needs about 10 time steps to reach the goal from the reset positions, which is located about 5cm above the foal position.

### 4.4. Learning from Demonstrations

Another method to incorporate prior information is to use demonstrations from an expert policy to guide exploration during RL. We first collected demonstrations with a keyboard controller. Then, we add a behavior cloning loss while performing RL that pushes the policy towards the demonstrator actions, as previously considered in (Nair et al., 2018). Instead of DDPG, the underlying algorithm RL algorithm used is TD3.

## 5. Experiments

We evaluate the industrial applicability of the residual RL approach on a variety of connector insertion tasks that are performed on a real robot, using easy-to-obtain reward signals. In this section, we consider two types of natural rewards which are intuitive to humans: an image directly specifying a goal and a binary sparse reward indicating success. For both cases, we report success rates on tasks they solve. We aim to answer the following questions: (1) Can such trained policies provide comparable performance to policies that are trained with densely-shaped rewards? (2) Are these trained policies robust to light variations and noise?

### 5.1. Vision-based Learning

For the vision-based learning experiments, we use only raw image observations and L1 distance in image space as the goal. Sample images that the robot received are shown in Fig. 6. We evaluate this type of reward on all three tasks. In our experiments, we use gray-scale images converted from RGB ones, this simplification reduces input space; but is good enough for our experiments.

### 5.2. Learning from Sparse Rewards

The applicability of a sparse reward function is explored on an insertion of the USB connector. The binary insertion signal is used as the metric for success. This experiment is most applicable to electronic manufacturing settings where the electrical connection between connectors can be directly measured.

### 5.3. Dense Reward Connector Plug Insertion

After evaluating the tasks in the above settings, we further evaluate with full state information with a dense and carefully shaped reward signal that incorporates distance to the goal and force information. Evaluating in this setting gives us an "oracle" that can be compared to the previous experiments in order to understand how much of a challenge sparse or image rewards pose for various algorithms.

### 5.4. Robustness

For safe and reliable future usage, it is required that the insertion controller is robust against small measurement or calibration errors that can occur when disassembling and reassembling a mechanical system. In this experiment, small goal perturbations are introduced in order to uncover the required setup precision of our algorithms.

### 5.5. Exploration Comparison

One advantage of using reinforcement learning is the exploratory behavior that allows the controller to adapt from new experiences unlike a deterministic control law. The two RL algorithms we consider in this paper, SAC and TD3, explore differently. SAC maintains a stochastic policy, and the algorithm also adapts the stochasticity through training. TD3 has a deterministic policy, but uses another noise process (in our case Gaussian) to inject exploratory behavior during training time. We compare the two algorithms, as well as when they are used in conjunction with residual RL, in order to evaluate the different exploration schemes.

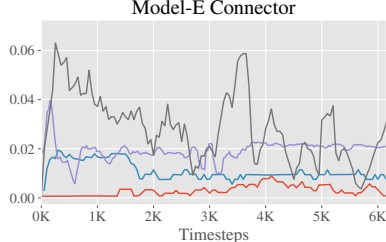
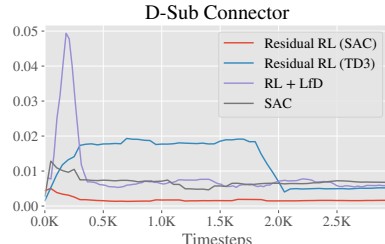

Figure 5. Resulting final mean distance during the vision-based training. The comparison includes residual RL, learning from demonstrations and pure RL, represented by SAC. Only residual RL with SAC manages to deal with the high-dimensional input and consistently solves all tasks after the given amount of training. The deterministic policies learn to move downwards, but often get stuck in the beginning of the insertion and fail to recover from unsuccessful attempts.

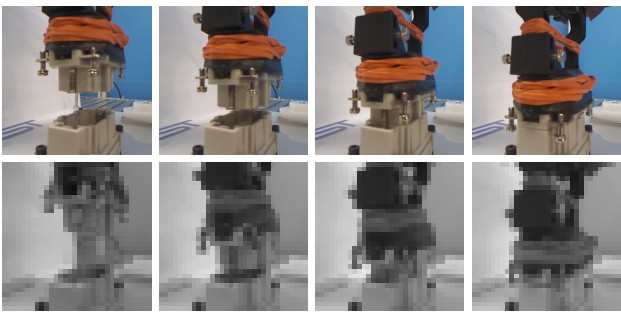

Figure 6. Successful insertion on the Module-E connector task. The 32 x 32 grayscale images are the only observations that the image-based reinforcement learning algorithm receives.

## 6. Results

The experimental results obtained in this work are described in the following.

### 6.1. Vision-based learning

The results of the vision-based experiment are shown in Figure 4. A successful and consistent insertion policy can be learned from relatively few samples, especially by Residual RL. This result suggests that goal-specification through images is a practical way to solve these types of industrial tasks. Sample images of successful insertion procedures are shown in Fig. 1.

Although image-based rewards are often very sparse and hard to learn from, in this case the distance between images corresponds to a relatively dense reward signal.

Interestingly, while learning from only reinforcement learning without residual RL, the policy would sometimes learn to "hack" the reward signal by moving down in the image in front of or behind the socket. In contrast, the stabilizing human-engineered controller provided sufficient horizontal control to transform the 3-dimensional task into a quasi

1-dimensional problem for the reinforcement learning algorithm.

### 6.2. Learning From Sparse Rewards

In this experiment, we compare several methods on the USB insertion task with sparse rewards. The results are reported in Fig. 7. First, we see that residual RL achieves a better final performance and requires less samples than RL alone, both in simulation and on physical hardware. Unlike residual RL, the pure RL approach needs to learn the structure of the position control problem from scratch, which explains the difference in sample efficiency. As samples are expensive to collect in the real world, residual RL is better suited for solving real-world tasks. Moreover, RL shows a broader spatial variance during training and needs to explore a wider set of states compared to residual RL, which can be potentially dangerous in hardware deployments.

Comparing the the final policies shows that the sparsely trained policy achieves the same performance after reasonably longer training. This result is especially surprising because the used actor critic algorithm generates a continuous value function over the whole state space domain. Non-differentiable, sparse rewards do not reveal the most rewarding direction to move, the reinforcement learning algorithm has to infer that from the given information. In fact, prior work has found that the final policy for sparse rewards can outperform the final policy for dense rewards as it does not suffer from a misspecified objective (Andrychowicz et al., 2017).

### 6.3. Dense Reward Connector Plug Insertion

The results of the experiment with dense rewards is shown in Fig. 4. In the dense rewards case, the same conclusions of residual RL outperforming SAC alone hold. Due to the shaped reward, SAC makes more initial progress, but cannot overcome the friction required to insert the plug in fully and complete the task.

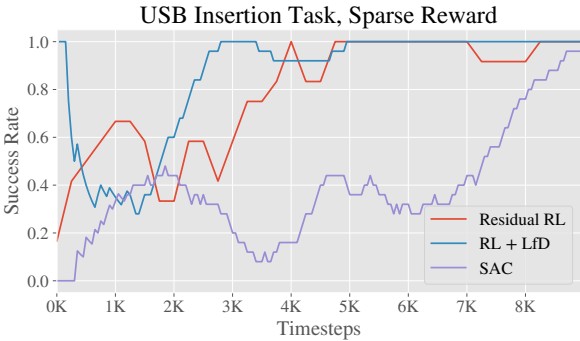

*Figure 7.* Learning curves for solving the USB insertion task with a sparse reward are shown. In this experiment, ground truth state is given as observations to the agent. Residual RL and RL with learning from demonstrations both solve the task relatively quickly, while RL alone (SAC) takes about twice as long to solve the task at the same level of performance.

## 6.4. Robustness

In previous set of experiments, the goal locations were known exactly. In this case, the hand-engineered controller performs well. However, once noise is induced to the goal location, the deterministic P-controller does not solve the task anymore. After training on perfect observations, a goal perturbation is created artificially and the controllers are tested under this condition. In the case of a $\pm 1$mm perturbation, the hand-engineered controller succeeding in only 15/25 trials, while residual RL still succeeds in 21/25 trials. These results are summarized in Tab. 1. In this experiment, the agent demonstrably learns consistent small corrective feedback behaviors in order to move in the right direction towards the descending path, a behavior that is very difficult to manually specify. The result of this experiment showcases the strength of residual RL. Since the human controller specifies the general trajectory of the optimal policy, environment samples are required only to learn this corrective feedback behavior.

## 6.5. Exploration Comparison

A comparison of TD3 and SAC is made in Fig. 8. When combined with residual RL, they perform comparably. When considering RL alone, TD3 learns the task faster than SAC. However, TD3 is significantly less robust, as shown in 1. These results are likely explained by the nature of the exploration for the two algorithms. TD3 has a deterministic policy and fixed noise during training, so once it observes some high-reward states, it quickly learns to repeat that trajectory. SAC requires more samples in order to adapt the noise to the correct scale, but once it does, the noise helps SAC stay robust to small peturbations. Furthermore, we found that the outputted action of TD3 approaches the extreme values at the edge of the allowed action space. This

suggests it finds a local minimum, which performs well, but may not be robust and TD3 cannot improve beyond that policy.

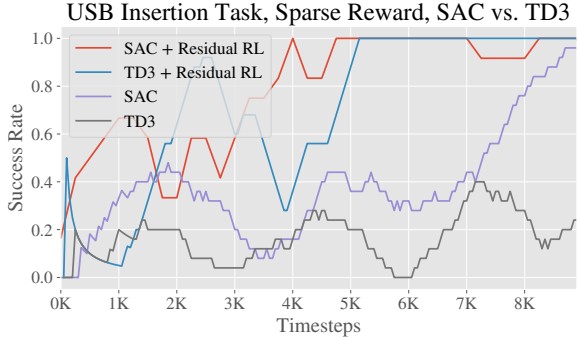

*Figure 8.* Comparison of two RL algorithms, SAC and TD3, on the USB insertion task with sparse rewards. Using residual RL, both algorithms can solve the task. Without residual RL, TD3 struggles to solve the task while SAC eventually does solve it to the same level of performance as the residual RL methods. We believe the difference may be due to the adaptive noise in SAC.

*Table 1.* Test-time performance on the USB insertion task. Noised is added in form of $\pm 1$ mm perturbations of the goal location. We report the average success rate out of 25 rollouts.

| USB | | Perfect Goal | Noisy Goal |
|---|---|---|---|
| Human Controller | | 100% | 60% |
| RL | | 16% | 8% |
| RL + LfD | | 100% | 32% |
| Residual RL | Dense | 100% | 84% |
| | Sparse, SAC | 88% | 84% |
| | Sparse, TD3 | 100% | 36% |
| | Images | 100% | 80% |

## 6.6. Requirements at precision and exploration

In order to explore the practical usability of controllers that are trained with deep reinforcement learning, we perform more experiments on other, industrially relevant, connectors. Residual RL policies are trained with dense rewards and with image-based observations and rewards for each of the adapters. The resulting success rates of out 25 rollouts are listed in Tab. III, no goal perturbation was applied. Comparing the residual RL policies with the pure version of its underlying P-controller yields that the inclusion of reinforcement learning significantly increases the success

*Table 2.* Performance evaluation on the D-Sub connector. We report the average success rate out of 25 rollouts of the trained policies.

| D-Sub connector | | Perfect Goal | Noisy Goal |
|---|---|---|---|
| Human Controller | | 100% | 44% |
| SAC | | 16% | 0% |
| Residual RL | Dense | 100% | 60% |
| | Images | 100% | 64% |

*Table 3.* Average success out of 25 policy executions on the Model-E connector. Pure RL, represented by SAC, did not manage to solve the task in the given amount of training.

| Model-E Connector | | Perfect Goal | Noisy Goal |
|---|---|---|---|
| Human Controller | | 52% | 24% |
| SAC | | 0% | 0% |
| Residual RL | Dense | 100% | 76% |
| | Images | 100% | 76% |

rate in tasks that are very difficult to model by hand. An example is the waterproof connector, which requires a very strong downwards pressing motion that needs to be executed at the right location. The human engineered controller was undergoing exhaustive tuning, but the robots control suffered from too much noise in order to precisely insert that connector in every trial.

The training on the different use cases is shown in Fig. 1. All the use cases showed great results with residual RL. Comparing the image-based training with the fully observed dense reward case shows that the image observations are sufficient to solve all the insertion tasks.

## 7. Discussion and Future Work

In this paper we studied residual RL with natural rewards and demonstrated that this approach can solve complex industrial assembly tasks with tight tolerances, e. g. connector plug insertions. We introduced vision inputs to the residual RL formulation, which increases the algorithm's usefulness for a wide range of industrial applications. Compared to previous work (Johannink et al., 2019), which uses dense reward signals, we showed that we can learn insertion policies only from sparse binary rewards or even purely from goal images. We conducted a series of experiments for various connector type assemblies and could demonstrate the feasibility of our method even under challenging conditions such as noisy goals and complex connector geometries. Our study motivates the application of residual RL to industrial

automation tasks, where reward shaping is not feasible, but sparse rewards or image goals can often be provided.

Future work will include more complex environments focusing on multi-stage assembly tasks through vision. This would pose a challenge to the goal-based policies as the background would be visually more complex. Moreover, multi-step tasks involve adapting to previous mistakes or inaccuracies, which could be difficult however, in theory should be able to be handled by RL. Extending the presented approach to multi-stage assembly tasks will pave the road to a higher robot autonomy in flexible manufacturing.

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
