# OpenReview forum: "Deep Reinforcement Learning for Industrial Insertion Tasks with Visual Inputs and Natural Reward Signals"
_ICML.cc/2019/Workshop/RL4RealLife — RL4RealLife 2019_

### Official Review · AnonReviewer2 · 2019-05-22
**Comments**

**Rating:** 3
**Confidence:** 4

**Review:**

This paper can be viewed as an application of the existing residual RL algorithm on connector insertion tasks. Probably the most significant result is that under imperfect sensing, residual RL performs better than a traditional P-controller, even if only sparse reward is available. The experiments are comprehensive. However, the results are not promising enough and the writing could be improved as certain statements can be quite confusing.

Pros:
1. Comprehensive experiments on various factors (e.g. connector type, state input, reward signals, prior knowledge, noise) essential to study of the task.

Cons:
1. Previous work on camera pose estimation (Yu et al, Siamese Convolutional Neural Network for Sub-millimeter-accurate Camera Pose Estimation and Visual Servoing) has shown that end-to-end training on a small static dataset would be enough to achieve good performance on similar tasks. Thus it seems that the major challenge comes from imperfect estimation of relative pose between end effector and objects. Probably the biggest concern here is whether residual RL is the right way to go, as the proposed method still needs access to the goal locations which may not be available in deployment. The performance is not comparable even if noisy goals are available.

2. There are certain writing issues that could cause confusion. For example, in the first sentence of 6.1, I believe it should be Figure 5 instead of "Figure 4". Also, the authors state in 6.6 that "no goal perturbation was applied" in experiments on other connectors. However, Tab. III contains entries for experiment result under "Noisy Goal". I would suggest dealing with these issues if paper gets accepted.

---

### Official Review · AnonReviewer1 · 2019-05-25
**Deep Reinforcement Learning for Industrial Insertion Tasks with Visual Inputs and Natural Reward Signals**

**Rating:** 4
**Confidence:** 3

**Review:**

Summary: The paper formulated the connector insertion tasks as the Markov decision process (MDP) and tried to use reinforcement learning methods to solve this problem. Furthermore, the paper tested the performance with two different reward specifications, sparse reward, and dense reward.

strengths.
(1)	The paper is well-written and readable.
(2)	The paper formulated the connector insertion tasks as the Markov decision process (MDP) and tested the performance on different reinforcement methods, TD3 and SAC.
(3)	The paper tests the performance of the reinforcement learning methods with two different reward specifications, sparse reward, and dense reward.
(4)	The performance of the residual RL demonstrated that RL methods were robust in the noise environment.

Weaknesses.
(1)	The paper lacks novelty. The reinforcement learning methods implemented in this paper were already tested in similar environments.

---

### Decision · Program_Chairs · 2019-05-28

Accept